# Guiding MCTS with Generalized Policies for Probabilistic Planning

**William Shen[1], Felipe Trevizan[1], Sam Toyer[2], Sylvie Thiébaux[1]** and **Lexing Xie[1]**

[1] The Australian National University  [2] University of California, Berkeley

[1]firstname.lastname@anu.edu.au [2]sdt@berkeley.edu

## Abstract

We examine techniques for combining generalized policies with search algorithms to exploit the strengths and overcome the weaknesses of each when solving probabilistic planning problems. The Action Schema Network (ASNet) is a recent contribution to planning that uses deep learning and neural networks to learn generalized policies for probabilistic planning problems. ASNets are well suited to problems where local knowledge of the environment can be exploited to improve performance, but may fail to generalize to problems they were not trained on. Monte-Carlo Tree Search (MCTS) is a forward-chaining state space search algorithm for optimal decision making which performs simulations to incrementally build a search tree and estimate the values of each state. Although MCTS can achieve state-of-the-art results when paired with domain-specific knowledge, without this knowledge, MCTS requires a large number of simulations in order to obtain reliable estimates in the search tree. By combining ASNets with MCTS, we are able to improve the capability of an ASNet to generalize beyond the distribution of problems it was trained on, as well as enhance the navigation of the search space by MCTS.

## 1  Introduction

Planning is the essential ability of a rational agent to solve the problem of choosing which actions to take in an environment to achieve a certain goal. This paper is mainly concerned with combining the advantages of forward-chaining state space search through UCT (Kocsis and Szepesvári 2006), an instance of Monte-Carlo Tree Search (MCTS) (Browne et al. 2012), with the domain-specific knowledge learned by Action Schema Networks (ASNets) (Toyer et al. 2018), a domain-independent learning algorithm. By combining UCT and ASNets, we hope to more effectively solve planning problems, and achieve the best of both worlds.

The Action Schema Network (ASNet) is a recent contribution in planning that uses deep learning and neural networks to learn generalized policies for planning problems. A generalized policy is a policy that can be applied to any problem from a given planning domain. Ideally, this generalized policy is able to reliably solve all problems in the

given domain, although this is not always feasible. ASNets are well suited to problems where "local knowledge of the environment can help to avoid certain traps" (Toyer et al. 2018). In such problems, an ASNet can significantly outperform traditional planners that use heuristic search. Moreover, a significant advantage of ASNets is that a network can be trained on a limited number of small problems, and generalize to problems of any size. However, an ASNet is not guaranteed to reliably solve all problems of a given domain. For example, an ASNet could fail to generalize to difficult problems that it was not trained on – an issue often encountered with machine learning algorithms. Moreover, the policy learned by an ASNet could be suboptimal due to a poor choice of hyperparameters that has led to an undertrained or overtrained network. Although our discussion is closely tied to ASNets, our contributions are more generally applicable to any method of learning a (generalized) policy.

Monte-Carlo Tree Search (MCTS) is a state-space search algorithm for optimal decision making which relies on performing Monte-Carlo simulations to build a search tree and estimate the values of each state (Browne et al. 2012). As we perform more and more of these simulations, the state estimates become more accurate. MCTS-based game-playing algorithms have often achieved state-of-the-art performance when paired with domain-specific knowledge, the most notable being AlphaGo (Silver et al. 2016). One significant limitation of vanilla MCTS is that we may require a large number of simulations in order to obtain reliable estimates in the search tree. Moreover, because simulations are random, the search may not be able to sense that certain branches of the tree will lead to sub-optimal outcomes. We are concerned with UCT, a variant of MCTS that balances the trade-off between exploration and exploitation. However, our work can be more generally used with other search algorithms.

Combining ASNets with UCT achieves three goals. (1) *Learn what we have not learned:* improve the capability of an ASNet to generalize beyond the distribution of problems it was trained on, and of UCT to bias the exploration of actions to those that an ASNet wishes to exploit. (2) *Improve on sub-optimal learning:* obtain reasonable evaluation-time performance even when an ASNet was trained with suboptimal hyperparameters, and allow UCT to converge to the optimal action in a smaller number of *trials*. (3) *Be robust to changes in the environment or domain:* improve perfor-

This paper is subsumed by "Guiding Search with Generalized Policies for Probabilistic Planning", which has been published in the Symposium on Combinatorial Search 2019.

mance when the test environment differs substantially from the training environment.

The rest of the paper is organized as follows. Section 2 formalizes probabilistic planning as solving a Stochastic Shortest Path problem and gives an overview of ASNets and MCTS along with its variants. Section 3 defines a framework for *Dynamic Programming UCT (DP-UCT)* (Keller and Helmert 2013). Next, Section 4 examines techniques for combining the policy learned by an ASNet with *DP-UCT*. Section 5 then presents and analyzes our results. Finally, Section 6 summarizes our contributions and discusses related and future work.

## 2  Background

A Stochastic Shortest Path problem (SSP) is a tuple $\langle S, s_0, G, A, P, C \rangle$ (Bertsekas and Tsitsiklis 1991) where $S$ is the finite set of states, $s_0 \in S$ is the initial state, $G \subseteq S$ is the finite set of goal states, $A$ is the finite set of actions, $P(s' \mid a, s)$ is the probability that we transition into $s'$ after applying action $a$ in state $s$, and $C(s, a) \in (0, \infty)$ is the cost of applying action $a$ in state $s$. A solution to an SSP is a stochastic policy $\pi \colon A \times S \to [0, 1]$, where $\pi(a \mid s)$ represents the probability action $a$ is applied in the current state $s$. An optimal policy $\pi^*$, is a policy that selects actions which minimize the expected cost of reaching a goal. For SSPs, there always exists an optimal policy that is deterministic which may be obtained by finding the fixed-point of the state-value function $V^*$ known as the Bellman optimality equation (Bertsekas and Tsitsiklis 1991), and the action-value function $Q^*$. That is, in the state $s$, we obtain $\pi^*$ by finding the action $a$ that minimizes $Q^*(s, a)$.

$$V^*(s) = \begin{cases} 0 & \text{if } s \in G \\ \min_{a \in A} Q^*(s, a) & \text{otherwise} \end{cases}$$

$$Q^*(s, a) = C(s, a) + \sum_{s' \in S} P(s' \mid a, s) \cdot V^*(s')$$

We handle dead ends using the finite-penalty approach (Kolobov, Mausam, and Weld 2012). That is, we introduce a fixed dead-end penalty $D \in (0, \infty)$ which acts as a limit to bound the maximum expected cost to reach a goal, and a *give-up* action which is selected if the expected cost is greater than or equal to $D$.

### 2.1  Action Schema Networks (ASNets)

The ASNet is a neural network architecture that exploits deep learning techniques in order to learn generalized policies for probabilistic planning problems (Toyer et al. 2018). An ASNet consists of alternating action layers and proposition layers (Figure 1), where the first and last layer are always action layers. The output of the final layer is a stochastic policy $\pi \colon A \times S \to [0, 1]$.

An action layer is composed of a single action module for each ground action in the planning problem. Similarly, a proposition layer is composed of a single proposition module for each ground proposition in the problem. These modules are sparsely connected, ensuring that only the relevant action modules in one layer are connected to a proposition

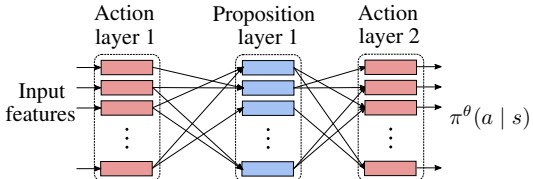

Figure 1: ASNet with 1 hidden layer (Toyer et al. 2018)

module in the next layer. An action module in one layer is connected to a proposition module in the next layer only if the ground proposition appears in the preconditions or effects of a ground action. Similarly, a proposition module in one layer is connected to an action module in the next layer only if the ground proposition appears in the preconditions or effects of the relevant ground action. Since all ground actions instantiated from the same action schema will have the same structure, we can share the same set of weights between their corresponding action modules in a single action layer. Similarly, weights are shared between proposition modules in a single proposition layer that correspond to the same predicate. It is easy to see that by learning a set of common weights $\theta$ for each action schema and predicate, we can scale an ASNet to any problem of the same domain.

ASNets only have a fixed number of layers, and are thus unable to solve all problems in domains that require arbitrarily long chains of reasoning about action–proposition relationships. Moreover, like most machine learning algorithms, an ASNet could fail to generalize to new problems if not trained properly. This could be due to a poor choice of hyperparameters, overfitting to the problems the network was trained on, or an unrepresentative training set.

### 2.2  Monte-Carlo Tree Search (MCTS)

MCTS is a state-space search algorithm that builds a search tree in an incremental manner by performing *trials* until we reach some computational budget (e.g. time, memory) at each decision step (Browne et al. 2012), at which point MCTS returns the action that gives the best estimated value.

A trial is composed of four phases. Firstly, in the selection phase, MCTS recursively selects nodes in the tree using a child selection policy until it encounters an unexpanded node, i.e. a node without any children. Next, in the expansion phase, one or more child nodes of the leaf node are created in the search tree according to the available actions. Now, in the simulation phase, a simulation of the scenario is played-out from one of the new child nodes until we reach a goal or dead end, or exceed the computational budget. Finally, in the backpropagation phase, the result of this trial is backpropagated through the selected nodes in the tree to update their estimated values. The updated estimates affect the child selection policy in future trials.

**Upper Confidence Bounds applied to Trees (UCT)** (Kocsis and Szepesvári 2006) is a variant of MCTS that addresses the trade-off between the exploration of nodes that have not been visited often, and the exploitation of nodes that currently have good state estimates. UCT treats

the choice of a child node as a multi-armed bandit problem by selecting the node which maximizes the Upper Confidence Bound 1 (UCB1) term, which we detail in the selection phase in Section 3.1.

**Trial-Based Heuristic Tree Search (THTS)**   (Keller and Helmert 2013) is an algorithmic framework that generalizes MCTS, dynamic programming, and heuristic search planning algorithms. In a THTS algorithm, we must specify five ingredients: action selection, backup function, heuristic function, outcome selection and the trial length. We discuss these ingredients and a modified version of THTS to additionally support UCT with ASNets in Section 3.

Using these ingredients, Keller and Helmert (2013) create three new algorithms, all of which provide superior theoretical properties over UCT: MaxUCT, Dynamic Programming UCT (DP-UCT) and UCT*. DP-UCT and its variant UCT*, which use *Bellman backups*, were found to outperform original UCT and MaxUCT. Because of this, we will focus on DP-UCT, which we formally define in the next section.

## 3   DP-UCT Framework

Our framework is a modification of DP-UCT from THTS. It is designed for SSPs with dead ends instead of finite horizon MDPs and is focused on minimizing the cost to a goal rather than maximizing rewards. It also introduces the *simulation function*, a generalization of random rollouts used in MCTS.

We adopt the representation of alternating decision nodes and chance nodes in our search tree, as seen in THTS. A decision node $n_d$ is a tuple $\langle s, C^k, V^k, \{n_1, \ldots, n_m\}\rangle$, where $s \in S$ is the state, $C^k \in \mathbb{Z}_0^+$ is the number of visits to the node in the first $k$ trials, $V^k \in \mathbb{R}_0^+$ is the state-value estimate based on the first $k$ trials, and $\{n_1, \ldots, n_m\}$ are the successor nodes (i.e. children) of $n_d$. A chance node $n_c$ is a tuple $\langle s, a, C^k, Q^k, \{n_1, \ldots, n_m\}\rangle$, where additionally, $a \in A$ is the action, and $Q^k$ is the action-value estimate based on the first $k$ trials.

We use $V^k(n_d)$ to refer to the state-value estimate of a decision node $n_d$, $a(n_c)$ to refer to the action of a chance node $n_c$, and so on for all the elements of $n_d$ and $n_c$. Additionally, we use $S(n)$ to represent the successor nodes $\{n_1, \ldots, n_m\}$ of a search node $n$, and we also employ the shorthand $P(n_d \mid n_c) = P(s(n_d) \mid a(n_c), s(n_c))$ and $c(n_c) = c(s(n_c), a(n_c))$. Initially, the search tree contains a single decision node $n_d$ with $s(n_d) = s_0$, representing the initial state of our problem.

### 3.1   Algorithm

UCT is described as an online planning algorithm, as it interleaves planning with execution. At each decision step, UCT returns an action either when a time cutoff is reached, or a maximum number of trials is performed. UCT then selects the chance node $n_c$ from the children of the root decision node that has the highest action-value estimate, $Q^k(n_c)$, and applies its action $a(n_c)$. We sample a decision node $n_d$ from $S(n_c)$ based on the transition probabilities $P(n_d \mid n_c)$ and set $n_d$ to be the new root of the tree.

A single trial under our framework consists of the selection, expansion, simulation and backup phase.

**Selection Phase.**   As described in THTS, in this phase we traverse the explicit nodes in the search tree by alternating between *action selection* for decision nodes, and *outcome selection* for chance nodes until we reach an unexpanded decision node $n_d$, which we call the tip node of the trial.

Action selection is concerned with selecting a child chance node $n_c$ from the successors $S(n_d)$ of a decision node $n_d$. UCT selects the child chance node that maximizes the UCB1 term, i.e. $\arg\max_{n_c \in S(n_d)} \text{UCB1}(n_d, n_c)$, where

$$\text{UCB1}(n_d, n_c) = \underbrace{B \cdot \sqrt{\frac{\log C^k(n_d)}{C^k(n_c)}}}_{\text{exploration}} - \underbrace{Q^k(n_c)}_{\text{exploitation}}.$$

$B$ is the bias term which allows us to adjust the trade-off between exploration and exploitation. We set $\text{UCB1}(n_d, n_c) = \infty$ if $C^k(n_c) = 0$ to force the exploration of chance nodes that have not been visited.

In outcome selection, we randomly sample an outcome of an action, i.e. sample a child decision node $n_d$ from the successors $S(n_c)$ of a chance node $n_c$ based on the transition probabilities $P(n_d \mid n_c)$.

**Expansion Phase.**   In this phase, we expand the tip node $n_d$ and optionally initialize the Q-values of its child chance nodes, $S(n_d)$. Calculating an estimated Q-value requires calculating a weighted sum of the form:

$$Q^k(n_c) = c(n_c) + \sum_{n_d \in S(n_c)} P(n_d \mid n_c) \cdot H(s(n_d)),$$

where $H$ is some domain-independent SSP heuristic function such as $h^{\text{add}}$, $h^{\text{max}}$, $h^{\text{pom}}$, or $h^{\text{roc}}$ (Teichteil-Königsbuch, Vidal, and Infantes 2011; Trevizan, Thiébaux, and Haslum 2017). This can be expensive when $n_c$ has many successor decision nodes.

**Simulation Phase.**   Immediately after the expansion phase, we transition to the simulation phase. Here we perform a simulation (also known as a rollout) of the planning problem from the tip node's state $s(n_d)$, until we reach a goal or dead-end state, or exceed the *trial length*. This stands in contrast to the behaviour of THTS, which lacks a simulation phase and would continuously switch between the selection and expansion phases until the *trial length* is reached.

We use the *simulation function* to choose which action to take in a given state, and sample the next state according to the transition probabilities. If we complete a simulation without reaching a goal or dead end, we add a heuristic estimate $H(s')$ to the rollout cost, where $s'$ is the final rollout state. If $s'$ is a dead end, then we set the rollout cost to be the dead-end penalty $D$.

The trial length bounds how many steps can be applied in the simulation phase, and hence allows us to adjust the lookahead capability of DP-UCT. By setting the trial length to be very small, we can focus the search on nodes closer to the root of the tree, much like breadth-first search (Keller and Helmert 2013). Following the steps above, if the trial length is 0, we do not perform any simulations and simply take a heuristic estimate for the tip node of the trial, or $D$ if the tip node represents a dead-end.

Traditional MCTS-based algorithms use a random simulation function, where each available action in the state has the same probability of being selected. However, this is not very suitable for SSPs as we can continuously loop around a set of states and never reach a goal state. Moreover, using a random simulation function requires an extremely large number of simulations to obtain good estimates for state-values and action-values within the search tree. Because of this, the simulation phase in MCTS-based algorithms for planning is often neglected and replaced by a heuristic estimate. This is equivalent to setting the trial length to be 0, where we backup a heuristic estimate once we expand the tip node of the trial.

However, there can be situations where the heuristic function is misleading or uninformative and thus misguides the search. In such a scenario, it could be more productive to use a random simulation function, or a simulation function influenced by domain-specific knowledge (i.e., the knowledge learned by an ASNet) to calculate estimates.

**Backup Phase.** After the simulation phase, we must propagate the information we have gained from the current trial back up the search tree. We use the *backup function* to update the state-value estimate $V^k(n_d)$ for decision nodes and the action-value estimate $Q^k(n_c)$ for chance nodes. We do this by propagating the information we gained during the simulation in reverse order through the nodes in the trial path, by continuously applying the backup function for each node until we reach the root node of the search tree.

Original UCT is defined with Monte-Carlo backups, in which the transition model is unknown and hence estimated based on the number of visits to nodes. However, in our work we consider the transition model to be known a priori. For that reason, DP-UCT only considers Bellman backups (Keller and Helmert 2013), which additionally take the probabilities of outcomes into consideration when backing up action value estimates $Q^k(n_c)$:

$$V^k(n_d) = \begin{cases} 0 & \text{if } s(n_d) \text{ is a goal} \\ D & \text{if } s(n_d) \text{ is a dead end} \\ \min_{n_c \in S(n_d)} Q^k(n_c) & \text{otherwise,} \end{cases}$$

$$Q^k(n_c) = \min \left\{ D, c(n_c) + \sum_{n_d \in \Upsilon^k(n_c)} \hat{P}(n_d \mid n_c) \cdot V^k(n_d) \right\},$$

$$\text{where } \Upsilon^k(n_c) = \left\{ n_d \mid n_d \in S(n_c), \, C^k(n_d) > 0 \right\},$$

$$\text{and } \hat{P}(n_d \mid n_c) = \frac{P(n_d \mid n_c)}{\sum_{n'_d \in \Upsilon^k(n_c)} P(n'_d \mid n_c)}.$$

$\Upsilon^k(n_c)$ represents the child decision nodes of $n_c$ that have already been visited in the first $k$ trials and hence have state-value estimates. Thus, $\hat{P}(n_d \mid n_c)$ allows us to weigh the state-value estimate $V^k(n_d)$ of each visited child decision node $n_d$ proportionally by its probability $P(n_d \mid n_c)$ and that of the unvisited child decision nodes.

It should be obvious that Bellman backups are derived directly from the Bellman optimality equations we presented in Section 2. Thus a flavor of UCT using Bellman backups is

asymptotically optimal given a correct selection of ingredients that will ensure all nodes are explored infinitely often.

## 4 Combining DP-UCT with ASNets

### 4.1 Using ASNets as a Simulation Function

Recall that an ASNet learns a stochastic policy $\pi \colon A \times S \to [0, 1]$, where $\pi(a \mid s)$ represents the probability action $a$ is applied in state $s$. We introduce two simulation functions which make use of a trained ASNet: STOCHASTIC AS-NETS which simply samples from the probability distribution given by $\pi$ to select an action, and MAXIMUM ASNETS which selects the action with the highest probability – i.e. $\arg\max_{a \in A(s)} \pi(a \mid s)$.

Since the navigation of the search space is heavily influenced by the state-value and action-value estimates we obtain from performing simulations, DP-UCT with an ASNet-based simulation function would ideally converge to the optimal policy in a smaller number of simulations compared to if we used the random simulation function. Of course, we expect this to be the case if an ASNet has learned some useful features or tricks about the environment or domain of the problem we are tackling.

However, using ASNets as a simulation function may not be very robust if the learned policy is misleading and uninformative. Here, robustness indicates how well UCT can recover from the misleading information it has been provided. In this situation, DP-UCT with ASNets as a simulation function would require a significantly larger number of simulations in order to converge to the optimal policy than DP-UCT with a random simulation function. Regardless the quality of the learned policy, DP-UCT remains asymptotically optimal when using an ASNet-based simulation function if the selection of ingredients guarantees that our search algorithm will explore all nodes infinitely often. Nonetheless, an ASNet-based simulation function should only be used if its simulation from the tip node $n_d$ better approximates $V^*(n_d)$ than a heuristic estimate $H(s(n_d))$.

**Choosing between STOCHASTIC ASNETS and MAXIMUM ASNETS.** We can perceive the probability distribution given by the policy $\pi$ of an ASNet to represent the 'confidence' the network has in applying each action. Obviously, MAXIMUM ASNETS will completely bias the simulations towards what an ASNet believes is the best action for a given state. If the probability distribution is highly skewed towards a single action, then MAXIMUM ASNETS would be the better choice, as the ASNet is very 'confident' in its decision to choose the corresponding action. On the other hand, if the probability distribution is relatively uniform, then STOCHASTIC ASNETS would likely be the better choice. In this situation, the ASNet may be uncertain and not very 'confident' in its decision to choose among a set of actions. Thus, to determine which ASNet-based simulation function to use, we should carefully consider to what extent an ASNet is able to solve the given problem reliably.

### 4.2 Using ASNets in UCB1

The UCB1 term allows us to balance the trade-off between exploration of actions in the search tree that have not been

applied often, and exploitation of actions that we already know have good action-value estimates based on previous trials. By including an ASNet's influence within UCB1 through its policy $\pi$, we hope to maintain this fundamental trade-off yet further bias the action selection to what the ASNet believes are promising actions.

**Simple ASNet Action Selection.** We select the child chance node $n_c$ of a decision node $n_d$ that maximizes:

$$\text{SIMPLE-ASNET}(n_d, n_c) = \frac{M \cdot \pi(n_c)}{C^k(n_c)} + \text{UCB1}(n_d, n_c)$$

$$= \underbrace{\frac{M \cdot \pi(n_c)}{C^k(n_c)} + B \cdot \sqrt{\frac{\log C^k(n_d)}{C^k(n_c)}}}_{exploration} - \underbrace{Q^k(n_c)}_{exploitation}$$

where $M \in \mathbb{R}^+$ and $\pi(n_c) = \pi(a(n_c) \,|\, s(n_c))$ for the stochastic policy $\pi$ learned by ASNet. Similar to UCB1, if a child chance node $n_c$ has not been visited before (i.e., $C^k(n_c) = 0$), we set $\text{SIMPLE-ASNET}(n_d, n_c) = \infty$ to force its exploration. The new parameter $M$, called the influence constant, allows us to control the exploitation of an ASNet's policy $\pi$ for exploration and, the higher $M$ is, the higher the influence of the ASNet in the action selection.

Notice that the influence of the ASNet diminishes as we apply the action $a(n_c)$ more often because $M \cdot \pi(n_c)/C^k(n_c)$ decreases as the number of visits to the chance node $n_c$ increases. Moreover, since the bias provided by $M \cdot \pi(n_c)/C^k(n_c)$ diminishes to 0 as $C^k(n_c) \rightarrow \infty$ faster than $B \cdot \sqrt{\log C^k(n_d)/C^k(n_c)}$ (i.e., the original UCB1 bias term), SIMPLE-ASNET preserves the asymptotic optimality of UCB1: as $C^k(n_c) \rightarrow \infty$, $\text{SIMPLE-ASNET}(n_d, n_c)$ equals $\text{UCB1}(n_d, n_c)$ and both converge to the optimal action-value $Q^*(n_c)$ (Kocsis and Szepesvári 2006).

Because of this similarity with UCB1 and their same initial condition (i.e., treating divisions by $C^k(n_c) = 0$ as $\infty$), we expect that SIMPLE-ASNET action selection will be robust to any misleading information provided by the policy of a trained ASNet. Nonetheless, the higher the value of the influence constant $M$, the more trials we require to combat any uninformative information.

**Ranked ASNet Action Selection.** One pitfall of the infinite exploration bonus in SIMPLE-ASNET action selection when $C^k(n_c) = 0$ is that all child chance nodes must be visited at least once before we actually exploit the policy learned by an ASNet. Ideally, we should use the knowledge learned by an ASNet to select the order in which unvisited chance nodes are explored. Thus, we introduce RANKED-ASNET action selection, an extension to SIMPLE-ASNET action selection.

$$\text{RANKED-ASNET}(n_d, n_c) =$$

$$\begin{cases} \text{SIMPLE-ASNET}(n_d, n_c) & \text{if } \forall n_c' \in S(n_d), C^k(n_c') > 0 \\ \pi(n_c) & \text{if } C^k(n_c) = 0 \\ -\infty & \text{otherwise} \end{cases}$$

The first condition stipulates that all chance nodes are selected and visited at least once before SIMPLE-ASNET ac-

tion selection is used. Otherwise, chance nodes that have already been visited are given a value of $-\infty$, while the values of unvisited nodes correspond to their probability in the policy $\pi$. Thus, unvisited child chance nodes are visited in decreasing order of their probability within the policy $\pi$.

RANKED-ASNET action selection will allow DP-UCT to focus the initial stages of its search on what an ASNet believes are the most promising parts of the state space. Given that the ASNet has learned some useful knowledge of which action to apply at each step, we expect RANKED-ASNET action selection to require a smaller number of trials to converge to the optimal action in comparison with SIMPLE-ASNET action selection. However, RANKED-ASNET may not be as robust as SIMPLE-ASNET when the policy learned by an ASNet is misleading or uninformative. For example, if the optimal action has the lowest probability among all actions in the ASNet policy and is hence explored last, then we would require an increased number of trials to converge to this optimum.

**Comparison with ASNet-based Simulation Functions.** DP-UCT with ASNet-influenced action selection is more robust to misleading information than DP-UCT with an ASNet-based simulation function. Since SIMPLE-ASNET and RANKED-ASNET action selection decreases the influence of a network as we apply an action it has suggested more frequently, we will eventually explore actions that may have a small probability in the policy learned by the ASNet but are in fact optimal. We would require a much larger number of trials to achieve this when using an ASNet-based simulation function, as the state-value and action-value estimates in the search tree would be directly derived from ASNet-based simulations.

# 5 Empirical Evaluation

## 5.1 Experimental Setup

All experiments were performed on an Amazon Web Services EC2 c5.4x large instance with 16 CPUs and 32GB of memory. Each experiment was limited to one CPU core with a maximum turbo clock speed of 3.5 GHz. No restrictions were placed on the amount of memory an experiment used.

**Considered Planners.** For our experiments, we consider two baseline planners: the original ASNets algorithm and *UCT\**. The latter is a variation of DP-UCT where the trial length is 0 while still using UCB1 to select actions, Bellman backups as the backup function, and no simulation function. UCT\* was chosen as a baseline because it outperforms original DP-UCT due to its stronger theoretical properties (Keller and Helmert 2013). We consider four parametrizations of our algorithms – namely, (i) Simple ASNets, (ii) Ranked ASNets, (iii) Stochastic ASNets, and (iv) Maximum ASNets – where: parametrizations (i) and (ii) are UCT\* using SIMPLE and RANKED-ASNET action selection, respectively; and parametrizations (iii) and (iv) are DP-UCT with a problem-dependent trial length using STOCHASTIC and MAXIMUM ASNETS as the simulation function, respectively.

**ASNet Configuration.** We use the same ASNet hyperparameters as described by Toyer et al. to train each network. Unless otherwise specified, we imposed a strict two hour time limit to train the network, though in most situations, the network finished training within one hour. All ASNets were trained using an LRTDP-based (Bonet and Geffner 2003) teacher that used LM-cut (Helmert and Domshlak 2009) as the heuristic to compute optimal policies. We only report the time taken to solve each problem for the final results for an ASNet, and hence do not include the training time.

**DP-UCT Configuration.** For all DP-UCT configurations we used $h^{\text{add}}$ (Bonet and Geffner 2001) as the heuristic function because it allowed DP-UCT to converge to a good solution in a reasonable time in our experiments, and set the UCB1 bias parameter $B$ to $\sqrt{2}$. For all problems with dead ends, we enabled Q-value initialization, as it helps us avoid selecting a chance node for exploration that may lead to a dead end. We did not enable this for problems without dead ends because estimating Q-values is computationally expensive, and not beneficial in comparison to the number of trials that could have been performed in the same time frame.

We gave all configurations a 10 second time cutoff to do trials and limited the maximum number of trials to 10,000 at each decision step to ensure fairness. Moreover, we set the dead-end penalty to be 500. We gave each planning round a maximum time of 1 hour, and a maximum of 100 execution steps. We ran 30 rounds per planner for each experiment.

## 5.2 Domains

**Stack Blocksworld.** Stack Blocksworld is a special case of the deterministic Blocksworld domain in which the goal is to stack $n$ blocks initially on the table into a single tower. We train an ASNet to unstack $n$ blocks from a single tower and put them all down on the table. Since the network has never learned how to stack blocks, it completely fails at stacking the $n$ blocks on the table into a single tower. A setting like this one—where the distributions of training and testing problems have non-overlapping support—represents a near-worst-case scenario for inductive learners like AS-Nets. In contrast, stacking blocks into a single tower is a relatively easy problem for UCT*. Our aim in this experiment is to show that DP-UCT can overcome the misleading information learned by ASNet policy. We train an ASNet on unstack problems with 2 to 10 blocks, and evaluate DP-UCT and ASNets on stack problems with 5 to 20 blocks.

**Exploding Blocksworld.** This domain is an extension of deterministic Blocksworld, and is featured in the International Probabilistic Planning Competitions (IPPC). In Exploding Blocksworld, putting down a block can detonate and destroy the block or the table it was put down on. Once a block or the table is exploded, we can no longer use it; therefore, this domain contains unavoidable dead ends. A good policy avoids placing a block down on the table or down on another block that is required for the goal state (if possible). It is very difficult for an ASNet to reliably solve Exploding Blocksworld problems as each problem could have its own 'trick' in order to avoid dead ends and reach the goal with minimal cost.

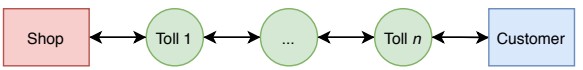

Figure 2: The CosaNostra Pizza Domain

We train an ASNet for 5 hours on a selected set of 16 problems (including those with avoidable and unavoidable dead ends) that were optimally solved by LRTDP within 2 minutes.[1] We evaluate ASNets and DP-UCT on the first eight problems from IPPC 2008 (Bryce and Buffet 2008). By combining DP-UCT and ASNets, we hope to exploit the limited knowledge and 'tricks' learned by an ASNet on the problems it was trained on to navigate the search space. That is, we aim to learn what we have not learned, and improve suboptimal learning.

**CosaNostra Pizza** (Toyer et al. 2018). The objective in CosaNostra Pizza is to safely deliver a pizza from the pizza shop to the waiting customer and then return to the shop. There is a series of toll booths on the two-way road between the pizza shop and the customer (Figure 2). At each toll booth, you can choose to either pay the toll operator or drive straight through without paying. We save a time step by driving straight through without paying but the operator becomes angry. Angry operators drop their toll gate on you and crush your car (leading to a dead end) with a probability of 50% when you next pass through their booth. Hence, the optimal policy is to only pay the toll operators on the trip to the customer, but not on the trip back to the pizza shop (as we will not revisit the booth). This ensures a safe return, as there will be no chance of a toll operator crushing your car at any stage. Thus, CosaNostra Pizza is an example of a problem with avoidable dead ends.

An ASNet is able to learn the trick of paying the toll operators only on the trip to the customer, and scales up to large instances while heuristic search planners based on determinisation (either for search or for heuristic computation) do not scale up (Toyer et al. 2018). The reason for the underperformance of determinisation-based techniques (e.g., using $h^{\text{add}}$ as heuristic) is the presence of avoidable dead ends in the CosaNostra domain. Moreover, heuristics based on delete relaxation (e.g., $h^{\text{add}}$) also underperform in the CosaNostra domain because they consider that the agent crosses each toll booth only once, i.e., this relaxation ignores the return path since it uses the same propositions as the path to the customer. Thus, we expect UCT* to not scale up to larger instances since it will require extremely long reasoning chains in order to always pay the toll operator on the trip to the customer; however, by combining DP-UCT with the optimal policy learned by an ASNet, we expect to scale up to much larger instances than UCT* alone.

For the CosaNostra Pizza problems, we train an ASNet on problems with 1 to 5 toll-booths, and evaluate DP-UCT and ASNets on problems with 2 to 15 toll booths.

---

[1]The training problems are available here: https://s3.amazonaws.com/ex-blocksworld/problems.zip

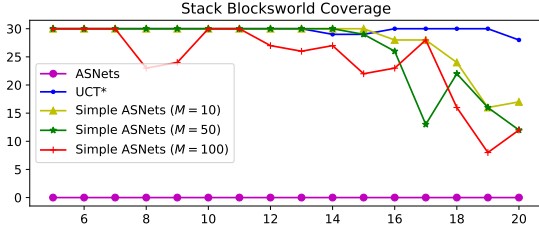

Figure 3: Coverage results for Stack Blocksworld.

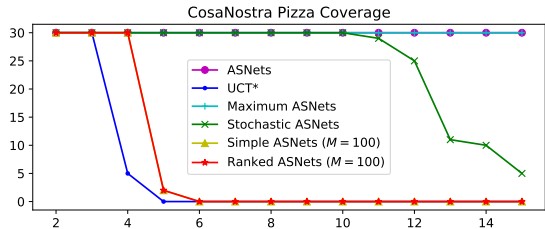

Figure 4: Coverage results for CosaNostra Pizza. Both AS-Nets and Maximum ASNets obtained perfect coverage.

## 5.3 Results

**Stack Blocksworld.** We allocate to each execution step $n/2$ seconds for all runs of DP-UCT, where $n$ is the number of blocks in the problem. We use Simple ASNets with the influence constant $M$ set to 10, 50 and 100 to demonstrate how DP-UCT can overcome the misleading information provided by the ASNet. We do not run experiments that use ASNets as a simulation function, as that would result in completely misleading state-value and action-value estimates in the search tree, meaning DP-UCT would achieve near-zero coverage.

Figure 3 depicts our results. ASNets achieves zero coverage, while UCT* is able to reliably achieve near-full coverage for all problems up to 20 blocks. In general, as we increase $M$, the coverage of Simple ASNets decays earlier as the number of blocks increases. This is not unexpected, as by increasing $M$, we increasingly 'push' the UCB1 term to select actions that the ASNet wishes to exploit, and hence misguide the navigation of the search space. Nevertheless, Simple ASNets is able to achieve near-full coverage for problems with up to 17 blocks for $M = 10$, 15 blocks for $M = 50$, and approximately 11 blocks for $M = 100$. We also observed a general increase in the time taken to reach a goal as we increased $M$, though this was not always the case due to the noise of DP-UCT.

This experiment shows that Simple ASNets is capable of *learning what ASNet has not learned* and being *robust to changes in the environment* by correcting the bad actions the ASNet suggests through search and eventually converging to the optimal solution.

**Exploding Blocksworld.** For all DP-UCT flavors, we increased the UCB1 bias parameter $B$ to 4 and set the maximum number of trials to 30,000 in order to promote more exploration. To combine DP-UCT with ASNets, we use Ranked ASNets with the influence constant $M$ set to 10, 50 and 100. Note, that the coverage for Exploding Blocksworld is an approximation of the true probability of reaching the goal. Since we only run each algorithm 30 times, the results are susceptible to chance.

Table 1 shows our results.[2] Since the training set used by ASNets was likely not representative of the evaluation problems (i.e., the IPPC 2008 problems), the policy learned by ASNets is suboptimal and failed to to reach the goal for the

---

[2]Since the difficulty of Exploding Blocksworld instances does not increase monotonically with problem size, presenting the results as a plot can be misleading.

relatively easy problems (e.g., p04 and p07) while UCT* was able to more reliably solve these problems.

By combining DP-UCT with ASNets through Ranked ASNets, we were able to either match the performance of UCT* or outperform it, even when ASNet achieved zero coverage for the given problem. However, for certain configurations, we were able to improve upon all other configurations. For p08, Ranked ASNets with $M = 50$ achieves a coverage of 10/30, while all other configurations of DP-UCT are only able to achieve a coverage of around 4/30. Despite the fact that the ASNet achieved zero coverage in this experiment, the general knowledge learned by the AS-Net helped us navigate the search tree more effectively and efficiently, even if the suggestions provided by the ASNet are not optimal. The same reasoning applies to the results for p04, where Ranked ASNets with $M = 50$ achieves a higher coverage than all other configurations.

We have demonstrated that we can exploit the policy learned by an ASNet to achieve more promising results than UCT* and the network itself, even if this policy is suboptimal. Thus, we have shown that Ranked ASNets is capable of *learning what the ASNet has not learned* and *improving the suboptimal policy* learned by the network.

**CosaNostra Pizza.** For this experiment, we considered ASNets as both a simulation function (Stochastic and Maximum ASNets), and in the UCB1 term for action selection (Simple and Ranked ASNets with $M = 100$) to improve upon UCT*. The optimal policy for CosaNostra Pizza takes $3n + 4$ steps, where $n$ is the number of toll booths in the problem. We set the trial length when using ASNets as a simulation function to be $\lfloor 1.25 \cdot (3n + 4) \rfloor$, where the 25% increase gives some leeway for Stochastic ASNets.

Figure 4 shows our results – the curves for ASNets and Maximum ASNets overlap, as well as the curves for Simple and Ranked ASNets. ASNets achieves full coverage for all problems, while UCT* alone is only able to achieve full coverage for the problems with 2 and 3 toll booths. Using ASNets in the action selection ingredient through Simple or Ranked ASNets with the influence constant $M = 100$ only allows us to additionally achieve full coverage for the problem with 4 toll booths. This is because Simple and Ranked ASNets guide the action selection towards the optimal action, but UCT still forces the exploration of other parts of the state space.

We are able to more reliably solve CosaNostra Pizza prob-

| Planner/Prob. | p01 | p02 | p03 | p04 | p05 | p06 | p07 | p08 |
|---|---|---|---|---|---|---|---|---|
| ASNets | 16/30
$8.0 \pm 0.0$
$0.18 \pm 0.14$s | 10/30
$12.0 \pm 0.0$
$0.17 \pm 0.01$s | 6/30
$10.0 \pm 0.0$
$0.2 \pm 0.04$s | - | 30/30
$6.0 \pm 0.0$
$0.19 \pm 0.07$s | 19/30
$12.0 \pm 0.0$
$0.42 \pm 0.12$s | - | - |
| UCT* | 26/30
$10.92 \pm 0.52$
$102.51 \pm 5.24$s | 9/30
$18.22 \pm 1.62$
$175.01 \pm 16.24$s | 13/30
$25.23 \pm 8.86$
$222.27 \pm 88.77$s | 11/30
$14.55 \pm 0.63$
$136.46 \pm 6.75$s | 30/30
$6.13 \pm 0.19$
$36.51 \pm 2.4$s | 28/30
$13.93 \pm 0.8$
$132.36 \pm 8.11$s | 30/30
$13.0 \pm 0.73$
$107.11 \pm 6.95$s | 5/30
$36.4 \pm 5.09$
$335.87 \pm 54.56$s |
| Ranked ASNets $M = 10$ | 25/30
$10.96 \pm 0.48$
$100.21 \pm 6.01$s | 6/30
$17.0 \pm 3.45$
$164.77 \pm 34.89$s | 11/30
$30.0 \pm 13.64$
$280.25 \pm 135.07$s | 10/30
$14.4 \pm 0.6$
$125.74 \pm 11.93$s | 30/30
$6.0 \pm 0.0$
$38.11 \pm 1.17$s | 25/30
$13.6 \pm 0.83$
$113.56 \pm 8.11$s | 30/30
$12.07 \pm 0.14$
$116.36 \pm 1.4$s | 4/30
$35.0 \pm 7.58$
$340.82 \pm 75.18$s |
| Ranked ASNets $M = 50$ | 23/30
$11.04 \pm 0.58$
$94.17 \pm 6.51$s | 10/30
$17.6 \pm 2.85$
$166.29 \pm 27.91$s | 14/30
$35.71 \pm 7.87$
$352.14 \pm 78.66$s | 15/30
$14.4 \pm 0.46$
$123.06 \pm 5.75$s | 30/30
$6.0 \pm 0.0$
$38.85 \pm 1.15$s | 27/30
$13.33 \pm 0.76$
$127.69 \pm 7.59$s | 30/30
$12.07 \pm 0.14$
$102.57 \pm 1.38$s | 10/30
$38.6 \pm 0.97$
$374.93 \pm 12.01$s |
| Ranked ASNets $M = 100$ | 25/30
$11.04 \pm 0.48$
$105.26 \pm 4.83$s | 12/30
$17.33 \pm 2.44$
$167.75 \pm 24.5$s | 14/30
$28.43 \pm 6.54$
$259.18 \pm 65.16$s | 10/30
$14.6 \pm 0.69$
$126.61 \pm 6.41$s | 30/30
$6.0 \pm 0.0$
$39.41 \pm 1.08$s | 29/30
$13.38 \pm 0.74$
$111.66 \pm 7.15$s | 30/30
$12.33 \pm 0.28$
$103.56 \pm 3.16$s | 4/30
$36.5 \pm 9.14$
$344.06 \pm 93.88$s |

Table 1: Results for Exploding Blocksworld. The coverage (i.e., the number of runs that successfully reached the goal) is presented in the 1st line of each cell. The 2nd and 3rd lines of each cell show the mean cost and mean time to reach a goal, respectively, and their associated 95% confidence interval.

lems when using ASNets as a simulation function. Since the ASNet learns the optimal policy, an ASNet-based simulation function allow us to obtain much better state-value estimates for nodes in the search tree than those provided by a domain-independent heuristic. It is easy to see that when we use Maximum ASNets, the state-value $V^*(n_d)$ for the tip node of a trial $n_d$ obtained from the simulation is optimal (assuming a sufficiently large trial length). Thus, Maximum ASNets achieves full coverage for all problems as Maximum ASNets will always provide DP-UCT with a path directly to the goal which it will eventually fall back to. For Stochastic ASNets, we see an exponential decay in the coverage as the problem size increases above 10 toll booths. The reason for this is because as the problem size increases, the probability of obtaining a path that leads directly to the goal decreases as the state space increases exponentially. Hence, DP-UCT cannot fall back to the path the ASNet has provided it, as this path may not have been taken before.

The explanations above also justify why Maximum AS-Nets took less time to reach a goal than all other configurations of DP-UCT. For this same reason, Maximum ASNets took less time to reach a goal than all other configurations of DP-UCT, e.g., for $n = 4$, the mean time to reach a goal and the 95% confidence interval for the considered planners are: ASNets ($0.15 \pm 0.05$s), UCT* ($64.95 \pm 7.16$s), Maximum ASNets ($54.51 \pm 0.19$s), Stochastic ASNets ($64.7 \pm 3.17$s), Simple ASNets with $M = 100$ ($104.45 \pm 2.38$s), Ranked ASNets with $M = 100$ ($124.41 \pm 7.27$s).

In this experiment, we have shown how using ASNets in UCB1 through SIMPLE-ASNET or RANKED-ASNET action selection can only provide marginal improvements over UCT* when the number of reachable states increases exponentially with the problem size, and the heuristic estimates are misleading. We also demonstrated how we can combat this *sub-optimal* performance of DP-UCT by using ASNets as a simulation function, as it allows us to more efficiently explore the search space and find the optimal actions. Thus, an ASNet-based simulation function may help DP-UCT *learn what it has not learned*.

**Triangle Tireworld** (Little and Thiébaux 2007). Triangle Tireworld is a domain with avoidable dead ends. ASNets is trivially able to find the optimal policy which always avoids dead ends. The results of our new algorithms on Triangle Tireworld are very similar to the results in the CosaNostra experiments, as the algorithms leverage the fact that ASNets finds the optimal generalized policy for both domains.

## 6 Conclusion, Related and Future Work

In this paper, we have investigated techniques to improve search using generalized policies. We discussed a framework for DP-UCT, extended from THTS, that allowed us to generate different flavors of DP-UCT including those that exploited the generalized policy learned by an ASNet. We then introduced methods of using this generalized policy in the simulation function, through STOCHASTIC ASNETS and MAXIMUM ASNETS. These allowed us to obtain more accurate state-value estimates and action-value estimates in the search tree. We also extended UCB1 to bias the navigation of the search space to the actions that an ASNet wants to exploit whilst maintaining the fundamental balance between exploration and exploitation, by introducing SIMPLE-ASNET and RANKED-ASNET action selection.

We have demonstrated through our experiments that our algorithms are capable of improving the capability of an ASNet to generalize beyond the distribution of problems it was trained on, as well as improve sub-optimal learning. By combining DP-UCT with ASNets, we are able to bias the exploration of actions to those that an ASNet wishes to exploit, and allow DP-UCT to converge to the optimal action in a smaller number of trials. Our experiments have also demonstrated that by harnessing the power of search, we may overcome any misleading information provided by an ASNet due to a change in the environment. Hence, we achieved the three following goals: (1) *Learn what we have not learned*, (2) *Improve on sub-optimal learning*, and (3) *Be robust to changes in the environment or domain*.

It is important to observe that our contributions are more generally applicable to any method of learning a (generalized) policy (not just ASNets), and potentially to other trial-

based search algorithms including (L)RTDP.

In the deterministic setting, there has been a long tradition of learning generalized policies and using them to guide heuristic Best First Search (BFS). For instance, Yoon et al. (Yoon, Fern, and Givan 2007) add the states resulting from selecting actions prescribed by the learned generalized policy to the the queue of a BFS guided by a relaxed-plan heuristic, and de la Rosa et al. (2011) learn and use generalized policies to generate lookahead states within a BFS guided by the FF heuristic. These authors observe that generalized policies provide effective search guidance, and that search helps correcting deficiencies in the learned policy. Search control knowledge à la TLPlan, Talplanner or SHOP2 has been successfully used to prune the search of probabilistic planners (Kuter and Nau 2005; Thiébaux et al. 2006). More recently, Steinmetz et al. (2016) have also experimented with the use of preferred actions in variants of RTDP (Barto, Bradtke, and Singh 1995) and AO* (Nilsson 1980), albeit with limited success. Our work differs from these approaches by focusing explicitly on MCTS as the search algorithm and, unlike existing work combining deep learning and MCTS (e.g. AlphaGo (Silver et al. 2016)), looks not only at using neural network policies as a simulation function for rollouts, but also as a means to bias the UCB1 action selection rule.

There are still many potential avenues for future work. We may investigate how to automatically learn the influence parameter $M$ for SIMPLE-ASNET and RANKED-ASNET action selection, or how to combat bad information provided by an ASNet in a simulation function by mixing ASNet simulations with random simulations. We may also investigate techniques to interleave planning with learning by using UCT with ASNets as a 'teacher' for training an ASNet, similar to the 'leapfrogging' idea presented by Groshev et al. (2018). ASNets may also be replaced by Deep Reactive Policies (Issakkimuthu, Fern, and Tadepalli 2018; Bajpai, Garg, and Mausam 2018), which learn reactive policies for RDDL problems. We hope that such work would bridge the gap between symbolic AI and deep learning, and improve the state-of-the-art in probabilistic planning.

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
