# OpenReview forum: "Guiding MCTS with Generalized Policies for Probabilistic Planning"
_icaps-conference.org/ICAPS/2019/Workshop/HSDIP_

### Official Review · AnonReviewer2 · 2019-04-05
**Very interesting work, but weak experimental results**

**Rating:** 9
**Confidence:** 5

**Review:**

The paper is well written, clearly structured and ASNets are an interesting framework to develop base policies. I think the paper will be an excellent one to have in the workshop.

I have also some criticism I hope will be perceived as constructive.

The authors read like being at pains to distance themselves from MCTS as a label... unfortunately, I think they algorithm is best described as an instance of MCTS. There is a fixed tree policy that looks ahead from the current state,  evaluates incrementally possible trajectories, and selects those that are judged to be more promising. Then at the leaves of the lookahead, a base policy is simulated to obtain an upper bound on the cost to go. Conceptually, DP-UCT + ASNets is pretty much like UCT + Random Walks.

I am not totally convinced about the experimental evaluation.  I would have expected the authors to compare on benchmarks we know DP-UCT or alternative algorithms, like Bonet and Geffner's Anytime AO*, to perform well. For instance, the Canadian Travelling Problem or some of the simpler domains of the latest IPPC.

That would allow to test whether ASNets can produce better cost to go estimates than the hand coded heuristics proposed by T. Keller and co-authors.

Also it would show  the results to generalises beyond domains like Exploding Blocks World, which in their original formulation do not make a great deal of sense. Working with RDDL has been difficult until recently, that libraries and tools made in Python to parsing and simulating execution have become available.

I think that if ASNets can match the performance of other MCTS algorithms that rely on domain-specific knowledge, or DRL algorithms like Value Iteration Networks, the authors' would have a very compelling demonstrator of their approach.

---

> ### Author Response · Authors · 2019-04-09
> **Thank you for your constructive feedback**
>
> We would like to thank Reviewer 2 for their positive feedback and constructive criticism. Our detailed replies may be found below.
>
> > “The authors read like being at pains to distance themselves from MCTS as a label... unfortunately, I think they algorithm is best described as an instance of MCTS.”
>
> Yes, we agree that all the algorithms we presented are instances of MCTS. We hope the reviewer understands that we did not intend to distance ourselves significantly from MCTS as a label. We noted that although our contributions are closely tied to UCT and ASNets, they may be adapted for use in other search algorithms (e.g., in a trial in (L)RTDP).
>
> > “I think they algorithm is best described as an instance of MCTS … Conceptually, DP-UCT + ASNets is pretty much like UCT + Random Walks.”
>
> The reviewer’s description of our algorithms is completely correct. Our framework allows us to construct instances of DP-UCT which are in themselves, instances of UCT which is an instance of MCTS. Indeed, DP-UCT + ASNets simulation function is very similar to UCT + Random Walks; the only difference is that the walks are guided by domain specific knowledge.
>
> > “Also it would show the results to generalises beyond domains like Exploding Blocks World, which in their original formulation do not make a great deal of sense.”
>
> Although Exploding Blocks World is far from a real world problem, it is a good benchmark for problems with unavoidable dead ends. For this reason we chose this domain.
>
> > Regarding experimental evaluation, comparison with other planning algorithms, RDDL, etc.
>
> Our initial investigation was to mainly show that (1) we can use search to improve upon the policy learned by an ASNet, even if this policy is uninformative or completely useless; (2) we can exploit the policy learned by an ASNet to improve the performance of the search algorithm when the policy is informative.  We agree that we should ultimately benchmark our contributions against comparable state-of-the-art planners, however our current implementation of DP-UCT is not computationally fast enough to compete with such planners.
>
> In regards to RDDL, ASNets are designed for PPDDL problems and are thus incompatible with RDDL. Moreover, the RDDL to PPDDL compilation generates PPDDL domains in which the actions mostly have no preconditions and several conditional effects. This structure is known to be hard for PPDDL planners and heuristics, and such is also true for ASNets.
>
> Nonetheless, ASNets could be replaced by ToRPIDo (Bajpai, Garg, Mausam) which learns transferrable policies (between problems) for RDDL, thus enabling our framework to be applicable to RDDL problems. We will include this in our future work section.
>
> References:
> Amulet Bajpai, Sankalp Garg, Mausam. Transfer of Deep Reactive Policies for MDP Planning. In Advances in Neural Information Processing Systems 31, NIPS 2018

---

> > ### Comment · AnonReviewer2 · 2019-04-09
> > **Interesting observation re PPDDL from RDDL**
> >
> > Thank you very much for your answers. I acknowledge your point re: positioning etc. excuse my overzealous reading.
> >
> > > In regards to RDDL, ASNets are designed for PPDDL problems and are thus incompatible with RDDL. Moreover, the RDDL to PPDDL compilation generates PPDDL domains in which the actions mostly have no preconditions and several conditional effects. This structure is known to be hard for PPDDL planners and heuristics, and such is also true for ASNets.
> >
> > This is an interesting observation regarding an experimental fact that was unknown to me. Since RDDL and PPDDL semantic relationship is murky, due to the (complicated) evolution of both languages, your remark seems to point out that there some kind of structural invariant that PPDDL encodings exploit implicitly that RDDL probably "obfuscates" by making it explicit. If that is the case, and in my opinion it would be good to study whether different compilations are possible that avoid doing that.
> >
> > RDDL representations are interesting because the language is familiar for researchers with backgrounds on control, stochastic optimisation or both. Literally, you are encoding an optimal control problem specifying constraints on inputs, stages (states) and trajectories (terminals and cost functions). That conceptual clarity I think is very valuable yet missing from PPDDL.
> >
> > The dichotomy between explicit (compilations to Mathematical Programming) and implicit methods (DP and approximations) is present in the literature both in planning and control. The gap in efficiency the authors describe seems to me to be expressing the dichotomy "in writing", so studying it may be productive and have far reaching consequences.
> >
> > Regarding benchmarks etc Exploding Blocksworld was not found to be a "probabilistically interesting" problem by the classic study due to Iain Smith and Sylvie Thiebaux, in proceedings of ICAPS 2007. Hence, unless the domain has been reformulated, I remain sceptical of its scientific value.

---

> > > ### Public Comment · ~Patrik_Haslum1 · 2019-04-10
> > > **Reference for the "probabilistically interesting" paper**
> > >
> > > I may be mistaken, but I think the paper you're referring to is "Probabilistic Planning vs Replanning". The authors are Iain Little and Sylvie Thiebaux. It appeared in the ICAPS 2007 ICAPS Workshop on the International Planning Competition: Past, Present and Future; as far as I know, no version of this has been published anywhere else. The workshop page seems to be dead, but there's a link to a copy of the paper from Sylvie's web page: http://users.cecs.anu.edu.au/~thiebaux/papers/icaps07wksp.pdf.
> > > I agree it's a classic paper, and very well worth a read.

---

> > > > ### Comment · AnonReviewer2 · 2019-04-11
> > > > **Thanks for correcting my memory**
> > > >
> > > > Cheers for the pointer, I couldn't find the ref because I was misremembering the name of the first author.

---

> > > ### Author Response · Authors · 2019-04-11
> > > **Exploding Blocks World**
> > >
> > > Regarding the exploding blocks worlds, it is a probabilistic interesting problem. Here is the quote from the paper "Probabilistic Planning vs Replanning" by Iain Little and Sylvie Thiébaux (p.7 column 2):
> > >
> > > "Exploding Blocksworld (...) The domain is probabilistically interesting and was in fact especially designed for a replanning strategy to perform poorly."
> > >
> > > We ran our experiments using the fixed version of the exploding blocks domain that prevents a block to be placed on top of itself, i.e., we fixed the flaw mentioned in the paper above before running our experiments.

---

> > > > ### Comment · AnonReviewer2 · 2019-04-11
> > > > **I was misremembering the details**
> > > >
> > > > Thanks for the correction, I was confused (and until Patrik didn't post the paper I couldn't find it because I was misremembering the surname of the first author). What stuck in memory was that FF-Replan was performing well in that domain, but as Little & Thiebaux remark in the sentences below yours:
> > > >
> > > > > The small 5-blocks problem instances selected, to which probabilistic
> > > > > planners can easily scale, are trivial. They are solved optimally by all the replanners we ?
> > > > > consider, including FF-replan, and by all IPC5 probabilistic planners.2 Moreover,
> > > > > probabilistic planners,
> > > > > unlike (suboptimal) deterministic ones, have trouble scaling
> > > > > to the next size (10 blocks), for which it is easy to randomly
> > > > > generate difficult instances. So from 10 blocks onwards, we
> > > > > have a situation were replanners are killed by their inability
> > > > > to do probabilistic reasoning, and probabilistic planners by
> > > > > their inability to scale.
> > > >
> > > > The finer point that FF-Replan was performing better than probabilistic planners due to the poor diversity in sizes in the benchmark used was washed away by time.
> > > >
> > > > The natural follow up question is then, what about Triangle Tireworld and the variants of it recently discussed by Geffner and Geffner in "Compact Policies for Fully Observable Non-Deterministic Planning as SAT", ICAPS 2018?

---

> > > > > ### Author Response · Authors · 2019-04-11
> > > > > **Other domains**
> > > > >
> > > > > Although triangle tire world is a probabilistic interesting problem, it is trivial for ASNets and ASNets is able to find a generalized policy that always avoids the dead ends (thus, the optimal policy for the triangle tire world problems). The results of our new algorithms on the triangle tire world would be very similar to the CosaNostra experiment since it can leverage the fact that ASNets find the optimal generalized policy for both domains.
> > > > >
> > > > > The advantage of testing our algorithms on the exploding blocks world domain is that it is a probabilistic interesting domain that is hard for ASNets. Moreover, ASNets managed to learn some structure of the domain but not enough to perform as well as UCT*. Thus, the exploding blocks world domain is a perfect test case for learning how to "correct" the ASNet policy (or similarly, learning when to follow the ASNet policy) in a challenging domain.

---

> > > > > > ### Comment · AnonReviewer2 · 2019-04-11
> > > > > > **Very interesting observation re: Tire World**
> > > > > >
> > > > > > I find very interesting that ASNets perform so well with triangle tireworld, and I will look into understanding why is that the case. Thanks very much for your feedback on this.

---

> ### Public Comment · ~Florian_Geißer1 · 2019-04-15
> **Question about RDDL python framework**
>
> I do not want to distract from the discussion of the paper, but I would appreciate if you could find some time to answer my question.
>
> > Also it would show  the results to generalises beyond domains like Exploding Blocks World, which in their original formulation do not make a great deal of sense. Working with RDDL has been difficult until recently, that libraries and tools made in Python to parsing and simulating execution have become available.
>
> I am not familiar with any python library for RDDL. Searching for some pointers online, the only related result I see is tf-rddlsim (https://github.com/thiagopbueno/tf-rddlsim). Is this the library you mean? If so, can you share your experiences with it, e.g. for using it in research or teaching?
>
> Thanks!

---

### Official Review · AnonReviewer1 · 2019-04-06
**Interesting idea, well-presented paper.**

**Rating:** 8
**Confidence:** 3

**Review:**

The paper introduces a new combination of MCTS with generalized policies for probabilistic planning. The generalized policies are based on the recently developed ASNets (Toyer et al. 2018), and are used in the simulation phase or in the action selection phase of UCT. The main idea of the paper is intuitive and its details are presented in a very clear manner. As mentioned by the authors, the rationale behind this combination of UCT with ASNets is to obtain the best of both worlds, exploiting during search the reactive knowledge learnt by the ASNets, but also overcoming through the search the potential weaknesses of the inductive learning approach embodied by theses ASNets. All of this is adequately analyzed and discussed in the experiments. Even though only three domains were tested, the experimental results and their explanations are sound and insightful. The paper concludes with a short overview of related research, together with some interesting ideas for future work.

The topic is well suited for the context of HSDIP; the idea is intuitive and clear; the presentation of the paper is well organized; the results are competitive with other methods based on
generalized policies for probabilistic planning, and are well analyzed. Besides all of these reasons, the topic of generalized planning has been drawing a good amount of attention recently, and I believe this paper could spark a nice discussion and provide some interesting future work ideas in the workshop, and would therefore argue for acceptance.


A few minor remarks:
 - Could you please put Figure 1 in the top of the column so it does not break
 the paragraph?

 - In Subsection 4.2, in the case-definition of Ranked-ASNet, the different cases are not mutually exclusive, i.e. it might happen that a node falls into both of the first two cases.

 - It could be good to provide references for RTDP, LRTDP, and AO*?

- It could be interesting to discuss the connection with the recent work by Issakkimuthu, Fern and Tadepalli (ICAPS 2018) about Deep Reactive Policies for probabilistic planning in the Future Work part.

---

> ### Author Response · Authors · 2019-04-09
> **Thank you for your review**
>
> We would like to thank Reviewer 1 for their positive feedback and useful remarks.
>
> We have addressed the remarks the reviewer made regarding Figure 1, and have fixed the cases in Ranked-ASNet to be mutually exclusive. We have also added references for RTDP, LRTDP and AO*.
>
> Issakkimuthu, Fern and Tadepalli’s work on Deep Reactive Policies (DRP) could replace ASNets in our framework. However, the DRPs they learn are problem dependent (as opposed to ASNets, which can generalize to all problems of a domain), so it is not clear if the cost of training a DRP would pay off. We will add this discussion to our related and future work.

---

### Meta-Review · Program_Chairs · 2019-04-25

**Recommendation:** Accept
**Confidence:** 5

**Metareview:**

Dear Authors,
thank you very much for your submission. We are happy to inform you that
we have decided to accept it and we look forward to your talk in the workshop.
Please, go over the feedback in the reviews and correct or update your papers
in time for the camera ready date (May 24).
Best regards
HSDIP organizers